# Interpretable Medical Image Classification with Self-Supervised Anatomical Embedding and Prior Knowledge

**Ke Yan**[1]                                           YANKE383@PAII-LABS.COM
**Youbao Tang**[1]                           TANGYOUBAO022@PAII-LABS.COM
**Adam P. Harrison**[1]               ADAMPHARRISON070@PAII-LABS.COM
**Jinzheng Cai**[1]                          CAIJINZHENG883@PAII-LABS.COM
**Le Lu**[1]                                             LE.LU@PAII-LABS.COM
[1] *PAII Inc., Bethesda, MD 20817, USA*

**Jingjing Lu**[2]                                CJR.LUJINGJING@VIP.163.COM
[2] *Beijing United Family Hospital, Beijing, 100015, PRC*

**Editors:** Under Review for MIDL 2021

## Abstract

In medical image analysis tasks, it is important to make machine learning models focus on correct anatomical locations, so as to improve interpretability and robustness of the model. We adopt a latest algorithm called self-supervised anatomical embedding (SAM) to locate point of interest (POI) on computed tomography (CT) scans. SAM can detect arbitrary POI with only one labeled sample needed. Then, we can extract targeted features from the POIs to train a simple prediction model guided by clinical prior knowledge. This approach mimics the practice of human radiologists, thus is interpretable, controllable, and robust. We illustrate our approach on the application of CT contrast phase classification and it outperforms an existing deep learning based method trained on the whole image.

**Keywords:** Interpretability, self-supervised, clinical knowledge, contrast phase, CT

## 1. Introduction

Deep convolutional neural networks (CNNs) have been widely used for medical image classification. However, their interpretability is often in question, which is critical for clinicians to trust the prediction results (Zhou and Greenspan et al., 2021). Specifically, it is important that the network's attention is on the correct body part of the image (Taylor et al., 2018). Network visualization methods have been used to address this problem (Zhou et al., 2019; Philbrick and Yoshida et al., 2018), but they are post-hoc and indirect. A better method is explicitly injecting clinical prior knowledge by specifying body parts of interest, and then guiding the network to learn from these regions. This can be achieved by training another segmentation or landmark detection model, but such a model needs numerous fine-grained training labels which are tedious to annotate.

Recently, Yan et al. proposed an algorithm called self-supervised anatomical embedding (SAM) (Yan et al., 2020), providing a potentially promising solution. SAM can be used to detect arbitrary anatomical locations in radiological images (CT, X-ray, etc.), meanwhile is completely unsupervised in training. In this paper, we study the task of contrast phase classification with the help of SAM. Intravenous contrast agents are routinely used in CT to increase vascular and tissue attenuation (Philbrick and Yoshida et al., 2018). Contrast phase classification is an important step for both training data curation and test data selection

when developing computer aided diagnosis algorithms (Zhou et al., 2019). The contrast information may be extracted from DICOM tags, but such hand-input tags are sometimes noisy and not standardized (Zhou et al., 2019). Radiologists identify the contrast phase of a CT by observing the intensity on specific anatomical structures such as aorta. We mimic radiologists' practice and use SAM to detect these landmarks, compute their Hounsfield unit (HU) values, and then train an SVM classifier for contrast phase classification. This method is simple yet effective.

## 2. Method and Data

SAM (Yan et al., 2020) is a novel pixel-level contrastive learning framework with a coarse-to-fine CNN and a hard negative sampling strategy. In an unsupervised manner, it learns to predict a global and a local embedding vector for each pixel in an image, so that the same anatomical location in different images express similar embeddings. SAM can be used to find correspondences between images efficiently and accurately. It has been employed in one-shot landmark detection in CT and X-ray (Yan et al., 2020).

Our task is to classify four phases in a dynamic liver CT as being non-contrast, arterial, venous, or delay (Zhou et al., 2019). From a hospital PACS, We collect 1,491 CT scans from 459 patients with text-mined phase labels. They are randomly divided to 1,000 training and 491 test samples without patient-level overlap. The phase labels of the test set are manually checked by a radiologist. The label distribution of the four classes are $290, 220, 297, 193$ in the training set and $127, 114, 142, 108$ in the test set.

To develop a phase classification model, a direct idea is to train a CNN using the 3D image as input (Zhou et al., 2019; Philbrick and Yoshida et al., 2018) and rely on the network to discover phase-related anatomical regions from training data. In contrast, our radiologist-inspired method explicitly designates the regions. We manually select eight anatomical points, including one in the heart, three in the aorta of the chest and abdomen, two in the hepatic portal vein, and two in the renal pelvis or ureter. According to the clinical prior knowledge, the HU values of these points correlate with contrast phases. First, we annotate the eight points on four scans (one scan per phase) of a random patient, i.e., the 3D coordinates of 32 points were recorded. Next, we compute the SAM embedding tensors of the four scans and extract the embedding vectors on these points as templates. Then, given another unlabeled CT, we compute its embedding tensors and use the 32 template vectors to convolve the tensor. The points with highest similarity scores are obtained as the matched anatomical points in the new image. On each point, we crop a $3 \times 3 \times 3$ image patch and adopt its maximum HU value as a feature, resulting in a 32D feature vector for each image. Finally, we train a simple linear SVM on the z-scored features of training set.

## 3. Experiment and Discussion

Table 1: Comparison of F1 scores for contrast phase classification, trained on 1,000 volumes and tested on 491 volumes.

| Method | Non-contrast | Arterial | Venous | Delay | Mean |
|---|---|---|---|---|---|
| 3DSE (Zhou et al., 2019) | **98.5** | **97.4** | 91.8 | 90.4 | 94.5 |
| SAM + SVM (proposed) | 98.4 | **97.4** | **94.4** | **93.4** | **95.9** |

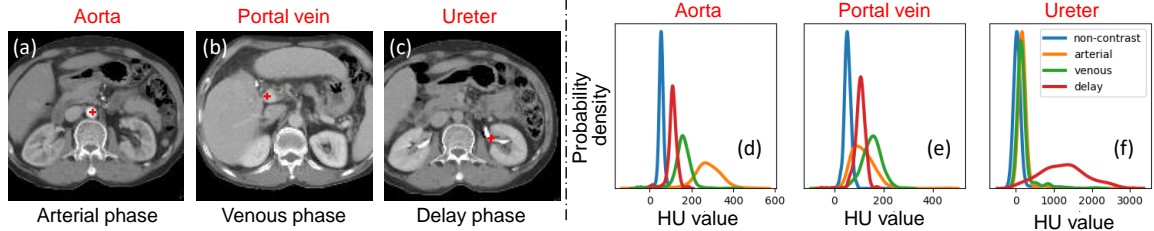

Figure 1: (a-c): Examples of phase-related anatomical landmarks detected by SAM. The phase label of each image is also shown. (d-f): The probability distribution of HU values of different phases on the three anatomical landmarks.

In this paper, we directly use the SAM model trained in (Yan et al., 2020). Figure 1 (a–c) displays examples of SAM accurately locating landmarks on new images based on our annotation on only one template study. With these landmarks detected on every image, we can compute the probability distribution of HU values on the whole dataset, see Figure 1 (d–f). The distribution is consistent with our prior knowledge: In aorta, the HU order is non-contrast < delay < venous < arterial; Venous is the brightest on hepatic vein while delay is the brightest on ureter. These HU values provide meaningful and interpretable features for phase classification. In Table 1, our method outperforms 3DSE (Zhou et al., 2019) especially on venous and delay phases. These two phases are relatively difficult to distinguish. Our method uses explicitly detected landmarks to learn a simpler yet better model, whereas the whole image classification method 3DSE (Zhou et al., 2019) has to learn important image regions from the scratch. The approach of combining SAM with other models using prior knowledge is expected to be applicable on more medical image tasks, e.g. disease classification on chest X-ray (Taylor et al., 2018).

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
