# OpenReview forum: "Interpretable Medical Image Classification with Self-Supervised Anatomical Embedding and Prior Knowledge"
_MIDL.io/2021/Conference/Short — MIDL 2021 Poster_

### Official Review · Reviewer_6tgD · 2021-04-29

**Confidence:** 3
**Final Rating:** 3

**Summary:**

This paper propose to encode prior knowledge for the interpretability of medical image classification. The specific task is to classify four phases in a dynamic liver CT as non-contrast, arterial, venous, or delay. To encode prior knowledge, they mannually select eight anatomical points, whose HU values correlate with contrast phase. Then they extract features on these points as templates to compute similarity with unlabeled image features to extract the representative features of the image, and finally train a simple linear SVM on the z-scored features. The experimental results show better performance than the previous method.

**Strengths:**

1. The interpretable classification task is well-motivated.
2. The proposed method only takes a small amout of additional annotation effort as the prior knowledge.
3. The proposed pipeline is light-weight.
4. The classification result outperforms the previous method.
5. The interpretation (HU value) seems to be consistent with prior knowledge.

**Weaknesses:**

1. How the HU value in Figure 1 is computed? What does the value mean? The authors need to provide more explanations, such as graphical computation process or equations for their method.
2. How is this paper comparing to other interpretable classification methods? As in the third paragraph of the 2. Method and Data:

To develop a phase classification model, a direct idea is to train a CNN using the 3D image as input (Zhou et al., 2019; Philbrick and Yoshida et al., 2018) and rely on the network to discover phase-related anatomical regions from training data.


**Deanonymize Review:**

no

**Justification Of The Rating:**

This paper propose an effective method for interpretation in classifying four phases in a dynamic liver CT and achieve good results. While as described in weaknesses, the presentation and comparison can be further improved.

**Paper Type:**

methodological development

**Special Issue:**

no

---

### Official Review · Reviewer_8CVt · 2021-05-01

**Confidence:** 4
**Final Rating:** 3

**Summary:**

The authors proposed applying self-supervised anatomical embedding (SAM) to improve the interpretability and robustness of the models in medical image analysis tasks. They prove the feasibility of the application by applying the approach to the CT contrast phase classification problem. Their approach outperforms a state-of-the-art deep learning-based method on contrast phase classification.

**Strengths:**

The idea of applying SAM into phase recognition is interesting, and it helps to classify the contrast phase of CT more efficiently.

The experience can prove the effectiveness of adopting SAM in the CT contrast phase classification.

The authors give enough review of prior work given the limited page.


**Weaknesses:**

The English writing should be improved, especially the usage of the articles.

The results, as they are presented, do not give me significant confidence in the relative performance of this method. The experience seems to be not adequate. The approach was tested on only one dataset and compare with only one method.

The author used the private dataset. Therefore, the experience result is not reproducible.


**Deanonymize Review:**

yes

**Justification Of The Rating:**

They proposed the adoption of SAM to improve the interpretability and robustness of the model. The idea is interesting, and they can prove the effectiveness of the proposed method through experience. However, the result is based on the private dataset; hence it is not reproducible.

**Paper Type:**

validation/application paper

**Special Issue:**

no

---

### Meta-Review · Area_Chair_YfSs · 2021-05-09

**Recommendation:** Accept (Poster)
**Confidence:** 4

**Metareview:**

While the writing of the paper requires improvement and the method is unfortunately not fully reproducible, both reviewers found interest in this simple yet robust and interpretable approach for contrast phase classification. I therefore recommend acceptance subject to improving the minor comments of both reviewers for a final version.

---

### Decision · Program_Chairs · 2021-05-11

Accept (Poster)